# Hybrid Orthorhombic Carbon Flakes Intercalated with Bimetallic Au-Ag Nanoclusters: Influence of Synthesis Parameters on Optical Properties

**DOI:** 10.3390/nano10071376

**Published:** 2020-07-15

**Authors:** Muhammad Abdullah Butt, Daria Mamonova, Yuri Petrov, Alexandra Proklova, Ilya Kritchenkov, Alina Manshina, Peter Banzer, Gerd Leuchs

**Affiliations:** 1Emeritus Group Leuchs, Max Planck Institute for the Science of Light, 91058 Erlangen, Germany; muhammad-abdullah.butt@mpl.mpg.de (M.A.B.); gerd.leuchs@mpl.mpg.de (G.L.); 2Institute of Optics, Information and Photonics, University Erlangen-Nuremberg, 91058 Erlangen, Germany; 3School of Advanced Optical Technologies, University Erlangen-Nuremberg, 91052 Erlangen, Germany; 4Institute of Chemistry, St. Petersburg State University, 198504 St. Petersburg, Russia; magwi@mail.ru (D.M.); proklova_97@mail.ru (A.P.); i.s.kritchenkov@spbu.ru (I.K.); 5Faculty of physics, St. Petersburg State University, 198504 St. Petersburg, Russia; y.petrov@spbu.ru

**Keywords:** laser-induced deposition, hybrid carbon-metal flake, orthorhombic carbon, metallic nanoparticles, polarization analysis

## Abstract

Until recently, planar carbonaceous structures such as graphene did not show any birefringence under normal incidence. In contrast, a recently reported novel orthorhombic carbonaceous structure with metal nanoparticle inclusions does show intrinsic birefringence, outperforming other natural orthorhombic crystalline materials. These flake-like structures self-assemble during a laser-induced growth process. In this article, we explore the potential of this novel material and the design freedom during production. We study in particular the dependence of the optical and geometrical properties of these hybrid carbon-metal flakes on the fabrication parameters. The influence of the laser irradiation time, concentration of the supramolecular complex in the solution, and an external electric field applied during the growth process are investigated. In all cases, the self-assembled metamaterial exhibits a strong linear birefringence in the visible spectral range, while the wavelength-dependent attenuation was found to hinge on the concentration of the supramolecular complex in the solution. By varying the fabrication parameters one can steer the shape and size of the flakes. This study provides a route towards fabrication of novel hybrid carbon-metal flakes with tailored optical and geometrical properties.

## 1. Introduction

In recent years, metamaterials and metasurfaces have paved the pathway for technological developments [1,2,3] from guiding and shaping light [4,5], enhanced optical effects [6,7], all the way to the design and fabrication of flat optics [8,9], countless applications have been suggested and realized. Corresponding metasurfaces are usually fabricated using electron-beam lithography [9,10], ion-beam milling, or chemical vapor deposition [11,12]. The key features of metamaterials and metasurfaces are ruled by the design of their building blocks. The most important components of metamaterials are metal nanostructures that are arranged in a regular or irregular fashion defining the optical response. In most cases reported to date, these nanostructures are either fabricated on dielectric-metal interfaces or embedded homogeneously into a dielectric environment. However, considering a crystalline and/or anisotropic embedding medium would be of interest as well, because it would add another parameter to the system enabling fine control of the optical properties. A special class of metamaterials is based on carbon nanostructures intercalated with metal clusters. They are considered as promising materials for optics, nanophotonics, electrocatalysis, and sensing technologies [13,14]. Most carbon-metal hybrid metamaterials are based on amorphous carbon [15,16]. Fullerenes (molecular form of carbon) and carbon nanotubes can also be decorated with metal NPs, rendering possible interesting applications in catalysis and sensing [17,18]. It should be noted that metamaterials based on regular 2D carbon structures (e.g., graphene), with plasmonic nanoparticles (NPs) on the surface, have led to the observation of interesting modulation effects of the plasmon response in the hybrid system caused by the influence of the host carbon matrix [19,20]. Therefore, the combination of metal NPs with crystalline carbon in a metamaterial is an exciting field. Graphene and thermally expanded graphite are the available options for this purpose. However, the first allows only surface decoration with NPs. Thermally expanded graphite in contrast can be intercalated with metal NPs, but it is of little use for optical studies due to the irregular morphology and structure [21]. The fabrication of novel and versatile crystalline carbon-metal metamaterials is thus of great interest for the field of optics and photonics. The progress in this direction is restrained by the difficulties in the synthesis of regular structures based on crystalline carbon and plasmonic NPs suitable for optical measurements.

A promising method for the fabrication of metal NPs embedded in a carbon matrix has been reported recently [22,23]. It is based on the decomposition of a dissolved supramolecular complex [Au_13_Ag_12_(C_2_Ph)_20_(PPh_2_(C_6_H_4_)_3_PPh_2_)_3_][PF_6_]_5_ (herein called SMC), as a result of laser irradiation of a substrate/solution interface. This leads to a controllable formation of a crystalline structure with Au-Ag nanoclusters embedded in the carbon matrix, on a substrate surface caused by the self-organization of the SMC constituents. Further details on SMC preparation and chemicals used can be found as the Appendix A. The laser-induced deposition of the hybrid carbon-metal structures is a straight-forward and easy-to-implement procedure that does not require intense laser irradiation or special equipment. This process results in the formation of NPs [23,24,25] or other structures such as flakes and flowers [26] of controllable composition, depending on the chosen SMC and the solvent. An interesting type of structures resulting from this laser-induced process are carbon flakes, which are the first reported orthorhombic form of carbon with embedded Au-Ag nanoclusters. Detailed analysis with a transmission electron microscope (TEM) revealed that the average radius of these metallic nanoclusters is d = 2R = 2.5 ± 0.9 nm, with a center to-center distance of 7.3 ± 1.5 nm [27]. Due to their inherent combination of bimetal Au-Ag nanoclusters with a crystalline carbon material of orthorhombic nature, these flakes possess intriguing optical properties [27], and may find applications in nano-optics and spectroscopy [7,22].

## 2. Experimental

### 2.1. Carbon Flakes Fabrication

In this article, we report on how various fabrication parameters affect the properties of the resulting hybrid carbon-metal flakes. This includes the laser irradiation time, the SMC concentration in the solution and the application of an external electric field during the growth process. Figure 1a shows the synthesis system that consists of a cuvette with the SMC solution mentioned above, prepared following a standard procedure [28], a substrate (industrial ITO-coated glass TIX 005 series from TECHINSTRO with a thickness of 1.1 mm, in contact with the solution) and a CW He-Cd Laser with λ = 325 nm and I = 0.1 W/cm^2^ (Plasma, JSC Research Institute of Gas Discharge Devices, Tsiolkovskz St., Rzayan, Russian Federation). The laser beam is illuminated on the substrate-solution interface from the side of the substrate.

The hybrid nanostructures are formed in the laser-affected zone on the substrate surface. The laser-induced deposition can be also realized with a DC electric field (7 V) applied via two metal electrodes (2 cm apart) deposited onto the ITO-coated glass (sample C3). The construction of the cuvette prevents the metal electrodes from being in contact with the SMC solution.

We experimentally investigate the effects of the aforementioned control parameters on the optical properties of the resulting carbon flakes. The investigated samples (containing multiple flakes each) with different fabrication parameters are listed in Table 1. Each of the parameters affects the reaction rate that all together lead to the variation of the flakes morphology and dimensions.

### 2.2. Methods

For the experimental study of the aforementioned carbon flakes, we use a custom method based on the microscopic Mueller matrix technique [26,27]. It allows performing a spatially resolved far-field polarization analysis of micron-sized specimens. The method combines the benefits of k-space microscopy [29,30] with liquid crystal variable retarders to avoid any moving optical elements in the analysis part of the setup [31].

For the optical investigation, we utilize a home-built setup [27] to raster-scan the carbon flakes with lateral dimensions between several micrometers to tens of micrometers, as shown in Figure 2. The incoming light beam of tunable wavelength and polarization is focused onto the sample using a microscope objective (Leica HCX PL FL 100×/0.9, Leica Microsystems GmbH, Wetzlar, Germany) of numerical aperture (NA) of 0.9 (effective NA = 0.5).

The small focal spot size (on the order of the wavelength) ensures that no edge effects contribute to the results. The light is collected using an oil immersion objective of NA 1.3 (Leica HCX PL FL 100×/1.3 Oil, Leica Microsystems GmbH, Wetzlar, Germany). The collected light passes through a set of two liquid crystal cells and a fixed polarizer to perform a complete Stokes analysis fully characterizing the polarization state of light [32]. The Stokes vector S can be calculated by projecting the incident light onto six polarization states (linear horizontal, vertical, diagonal, anti-diagonal as well as right- and left-handed circular).
(1)$$S={[s_{0}\;s_{1}\;s_{2}\;s_{3}]}^{T}$$
where $s_{0}=I_{h}+I_{v}$, $s_{1}=I_{h}-I_{v}$, $s_{2}=I_{p45}-I_{m45}$, $s_{3}=I_{RCP}-I_{LCP}$.

For polarimetry we use liquid crystal variable retarders (LCVR) from Thorlabs GmbH, 85232 Bergkirchen, Germany [33]. The retardance of these LCVRs can be controlled via a voltage applied across the cells. This property makes them suitable for the flexible polarization analysis [27,31]. In our technique, the six different polarization states are generated by aligning the optic axis of the liquid crystals at 22.5 and 45°, and by using a combination of different retardances (applied voltages). This way, all six polarization states can be projected onto a linear fixed polarizer. An additional lens is used to image the back focal plane of the lower objective onto a CCD camera with 8-bit dynamic range (DMK 31BU03, The Imaging Source Europe GmbH, Bremen, Germany). This allows us to access momentum space (k-space; see Figure 2b). Incoming and outgoing Stokes vectors are linked via a 4 × 4 matrix, the Mueller matrix. By performing a pseudo-inverse analysis of input and output Stokes vectors, we can extract the Mueller matrix in the following way [34].
(2)$$S_{out}=MS_{in}$$
(3)$$M=S_{out}\times S_{in}{}^{+}=S_{out}\times {\left(S_{in}{}^{†}S_{in}\right)}^{-1}S_{in}{}^{†}$$
where $S_{in}{}^{+}$ denotes the pseudo-inverse of $S_{in}$, which can be calculated using $S_{in}{}^{†}$, the conjugate transpose of $S_{in}$. The Mueller matrix contains information about the optical properties of the carbon flake, such as attenuation, diattenuation, and birefringence, which can be further extracted by using decomposition techniques [34,35,36].

## 3. Results and Discussion

We perform a computational analysis of the experimentally recorded data to extract the Mueller matrix of each flake investigated. Here, we analyze two optical properties of carbon flakes. Firstly, the attenuation of the flakes is studied. To compare the results for flakes of different thickness, we normalize the attenuation accordingly (attenuation per 100 nm thickness of the flake). Secondly, we study the birefringence Δn, which is related to the retardance *δ* at a particular wavelength *λ* and for a certain thickness *d* of the carbon flake in the following way:(4)$$\Delta n=\frac{\delta \lambda }{2\pi d}$$

The thicknesses of the studied flakes were determined by atomic force microscopy (AFM), while the lateral dimensions were analyzed in a scanning electron microscope (SEM). A spatial average across data in the central region of the flake is used to analyze the average value of the aforementioned optical properties. For the three fabrication parameters discussed in this report, we retrieved the following results.

SEM images of the flakes deposited at different experimental parameters are presented in Figure 1b–g (for sample C2, C3, C4, and C5). For all cases, minor fluctuations in the flake’s dimensions are observed. However, we can conclude that the dimensions of the flakes are clearly correlated with the deposition conditions. For instance, if a DC electric field is applied across the cuvette during the fabrication process (see Figure 1(f,g-C3)), the lateral dimensions of the flake structures are modified strongly in comparison to the case with the field switched off (see Figure 1(b,e-C2)). An elongation along one axis of the flakes is observed. The typical sizes of such flakes reach 40–60 µm in length and 5–10 µm in width (see Figure 1(f,g-C3)). These results suggest that the lateral dimensions of flakes can be fine-tuned by an externally applied voltage. To single out this effect only the electric field was switched on and off for samples C3 and C2, respectively, while all other parameters were kept unchanged. However, the optical properties, i.e., attenuation (normalized to the thickness) and birefringence are not significantly modified by this fabrication condition (see Figure 3).

Furthermore, the laser irradiation time was found to also directly influence the overall growth of the flakes. For samples C2 and C4, only the irradiation time was chosen to be different, keeping all other parameters the same. AFM and SEM scans of the carbon flakes for samples reveal a direct relation between the irradiation time and the thickness of the resulting flakes, with a longer irradiation resulting in thicker flakes. The results of the optical measurements presented in Figure 3 show unambiguously that the attenuation (normalized to the thickness) and birefringence are not significantly changed for different laser irradiation times.

By changing the third parameter studied here, the SMC concentration in the solution, we see clearly in Figure 3 that the attenuation of the individual flakes can be modified. We analyze sample C2 and C5, for which other fabrication parameters were kept constant and only the SMC concentration was changed. We observe a clear change in the attenuation for different SMC concentrations. To serve as a reference, we also plot the attenuation and birefringence (C1-Flake 1 and C1-Flake2) from [27] in Figure 3.

It is worth noting that the linear birefringence remains unchanged for all different SMC densities tested here. In addition, the SMC concentration was also found to affect the number of flakes formed on the substrate (not shown here). The structural parameters of the carbon flakes are found to not be effected by the SMC concentration (see Figure 1d-C5,e-C2).

In the theoretical modeling discussed in detail in our earlier study [27], we found that the attenuation depends sensitively on the embedded bimetallic nanoclusters whereas the linear birefringence is a direct consequence of the orthorhombic carbon lattice. Here, we confirm this result by analyzing the results for flakes fabricated with different parameters, which do not affect the crystalline phase of carbon and, hence, the birefringence does not change significantly. Therefore, we speculate that a lower density per unit volume of bimetallic nanoclusters embedded in the carbon flakes is the cause of the substantially lower attenuation observed for a lower SMC concentration. Some other parameters such as the choice of a substrate, electro-optical effects within the flakes, etc., not discussed here in more detail, could also affect the attenuation of flakes. We summarize in Table 2, all observed dependences of the tested fabrication parameters on the geometrical and optical properties of carbon flakes.

## 4. Conclusions

In conclusion, we studied the effect of different concentrations of the SMC in the solution, laser irradiation times, as well as the application of an external DC electric field during the fabrication process. The attenuation of the studied carbon flakes was found to be mainly affected by the SMC concentration. Applying an electric field results in lateral elongation of carbon flakes. The laser irradiation time directly influences the thickness of the resulting carbon flakes. In all cases, the birefringence was found to be wavelength independent and nearly constant around Δ*n* ≈ 0.1. Our study shows that the geometrical parameters of orthorhombic metal-carbon hybrid flakes can be tailored during the fabrication process by controlling various fabrication parameters, such as irradiation time and the application of an external field along the substrate. Furthermore, optical parameters such as the attenuation can be modified as well, while not changing the extraordinarily high linear birefringence of the flakes, which results from the orthorhombic carbon matrix. Hence, the material platform is promising to provide unique hybrid flakes, which can be engineered towards certain applications in opto-electronics and nano-optics.

## Figures and Tables

**Figure 1 nanomaterials-10-01376-f001:**
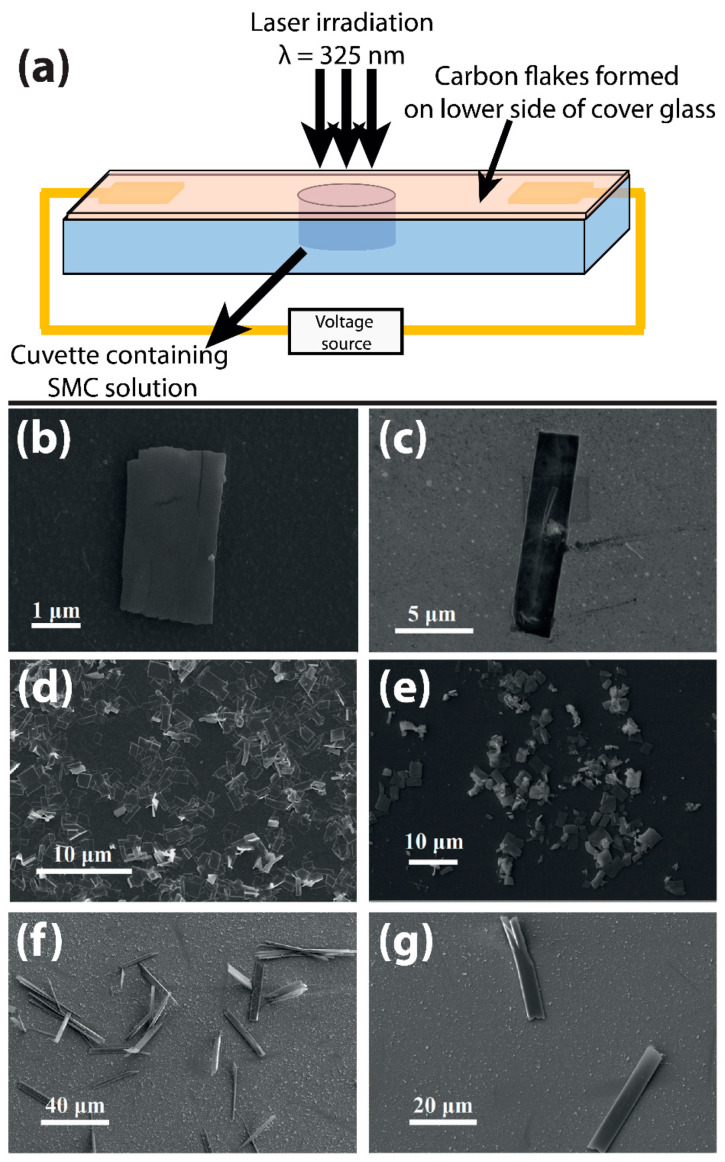
Fabrication of carbon flakes in dependence on various fabrication parameters. The scanning electron micrographs are from samples mentioned in Table 1. (**a**) The sample present in the cuvette with a certain concentration of supramolecular complex (SMC) in the solution is irradiated with laser for a certain amount of time. There is also the possibility to apply an electric field across the substrate to affect the formation of flakes. Flakes are formed by varying the laser irradiation time: (**b**) Sample C-2 with a laser irradiation time of 40 min and (**c**) sample C-4 with a laser irradiation time of 80 min. Varying SMC concentration: (**d**) Sample C-5 with an SMC concentration of 6 g/L and (**e**) sample C2 with an SMC concentration of 2 g/L. Images in (**f**) and (**g**) show examples of flakes from sample C-3 resulting from a fabrication with an external electric field switched on and causing lateral elongation of flakes.

**Figure 2 nanomaterials-10-01376-f002:**
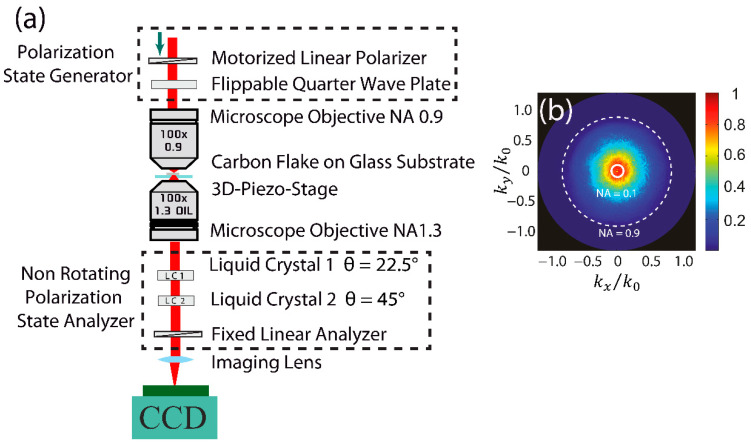
Experimental setup for the Mueller matrix measurement. (**a**) The polarization of the incident laser beam can be controlled (H, V, +45, −45, right circular polarization (RCP), and left circular polarization (LCP) with a motorized polarizer and a quarter-wave plate. The incident beam is focused onto the sample (flakes on the glass substrate), which is placed on a piezo stage to raster-scan the flakes with respect to the fixed input beam. The transmitted light is collected with another objective and polarization analysis is performed using liquid crystals. The back focal plane of the lower objective is imaged onto the charge-coupled device (CCD) camera and recorded for further analysis. (**b**) During the analysis of the Fourier space image, only k-vectors corresponding to normal incidence (numerical aperture (NA) region 0.1) are used for extracting the Mueller matrix.

**Figure 3 nanomaterials-10-01376-f003:**
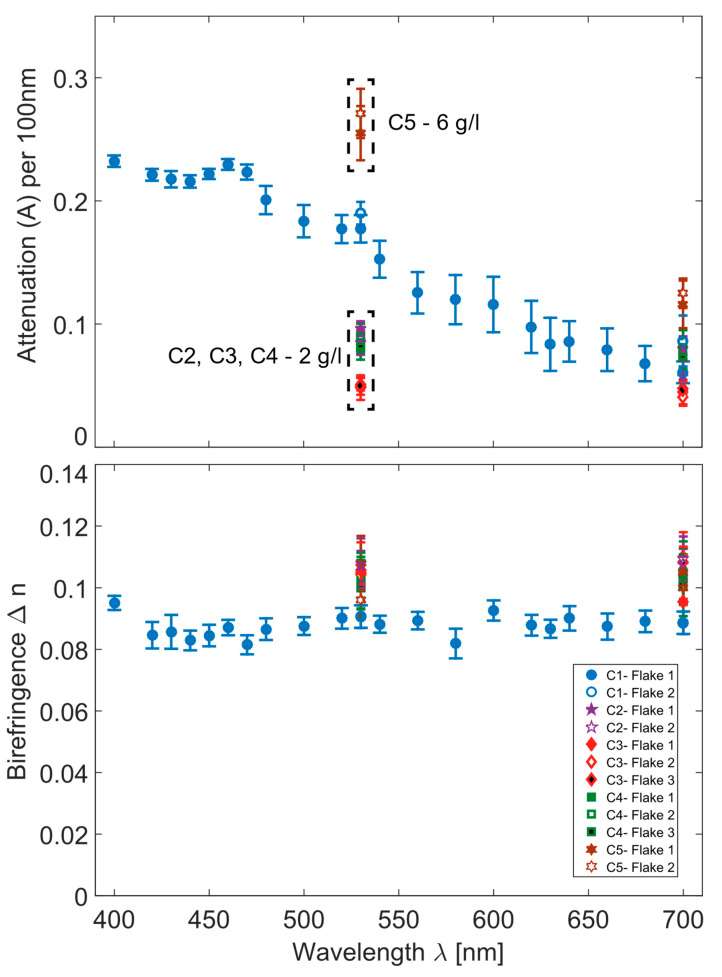
Experimental results for various flakes on different samples. The graphs contain data from our previous work [27] to serve as a reference (C1-Flake 1 and C1-Flake 2). The birefringence for flakes with different fabrications remain constant around Δn = 0.1. The attenuation is affected by the concentration of the SMC used during the fabrication process.

**Table 1 nanomaterials-10-01376-t001:** Samples with different fabrication parameters.

Sample	Laser Irradiation Time (min)	SMC Concentration (g/L)	Electric Field (V/m)	Resulting Thickness of Carbon Flakes (nm)
C1 [27]	15	4 *	Off	150–500
C2	40	2	Off	250–750
C3	40	2	On	250–750
C4	80	2	Off	1200–1700
C5	40	6	Off	150–750

* In the earlier experiments, additional phenylacetylene was present in the SMC solution. We did not yet study the effect of phenylacetylene systematically, but that preliminary results indicate that phenylacetylene does not change the composition of the generated flakes, but only changes the number of flakes formed.

**Table 2 nanomaterials-10-01376-t002:** Variable fabrication parameters and their effect on carbon flakes.

Varied Fabrication Parameter	Effect on Optical Properties of Carbon Flakes	Effect on Geometrical Properties of Carbon Flakes
Electric field on or off	Optical properties remain unchanged	Lateral elongation of flakes (with field switched on)
Laser irradiation time	Optical properties remain unchanged	Increase in thickness with increasing irradiation time
SMC concentration	Increase in attenuation with the increasing SMC concentration	No observed effect on structural parameters

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
