# Peer review of "Hybrid Orthorhombic Carbon Flakes Intercalated with Bimetallic Au-Ag Nanoclusters: Influence of Synthesis Parameters on Optical Properties"

_nanomaterials, 2020, doi:10.3390/nano10071376_

Round 1
Reviewer 1 Report
This paper described the synthesis of various types of hybrid orthorhombic carbon flakes by changing some synthetic parameters. The authors found that the shape, thickness as well as optical properties can be tuned by changing the synthetic parameters. This would be interesting for the readers of this journal.
However, this reviewer would like to point out that more details of the synthetic conditions should be added in this manuscript. For instance, the authors claimed that Au-Ag nanoclusters and SMC were used to synthesized the carbon flakes, however, no detailed description about them are included in this manuscript. In addition, details about the cover glass including the thickness as well as materials should also be added.
Additionally, this reviewer wondered that the size and the number of metal of the Au-Ag nanocluster may affect the structural parameters of the obtained carbon flakes. This reviewer suggest that the different type of Au-Ag nanocluster should be used to clarify the effect of nanocluster.
Reviewer 2 Report
The paper report about the optical properties of carbon flakes grown by laser irradiation under different experimental conditions.
The results shown that the different growth conditions affect the attenuation per 100 nm while birifrangence is almost unaffected.
I have some critics to the paper:
1) Authors should give a short description of the SMC solution preparation togehter with the used chemicals, in the experimental section.
2) Figure 1 report the SEM images of the samples grown under different experimental conditions. I suggest to report in the caption the samples ID as reported in table I. I identified : (b)-C2; (c)-C4; (d)-C5; (e)-C2; (f)-C2; (g)-C3. In this way the reader will not go forth and back from Figure 1 to Table 1 in order to identify the differences in the preparation.
3) page 5-149 : same as 2) e.g. '(see Figure 1f,g - C2,C3)' or something similar.
4) Figure 3: Report clearly which data come from ref. [27] in the figure or in the caption.
5) page 8-184: This is the first time that embedded NPs are referenced in the paper. It is not clear if the samples contain embedded NPs, and if so which are the dimension, nature etc. ...? Probably past works, referenced in the bibliography, report this information (see also point 2) , but I think that referenced past works should contain additional information but not all the needed ones. From this point of view the work seems to report some side results from past published works.
6) The last point regards the dimensions of the samples that exceed the nano scale so that probably Nanomaterials is not the appropriate journal where to publish the results.
Round 2
Reviewer 2 Report
I am satisfied with the authors' replies.